

# ChinaRiceCalendar-Seasonal Crop Calendars for Early, Middle, and Late Rice in China

Li Hui[1], Wang Xiaobo[2], Wang Shaoqiang[1,2,3,4], Liu Yuanyuan[2], Liu Zhenhai[2], Chen Shiliang[1,2], Wang Qinyi[1], Zhu Tongtong[1], Wang Lunche[1], Wang Lizhe[5]

[1]Key Laboratory of Regional Ecology and Environmental Change, School of Geography and Information Engineering, China University of Geosciences, Wuhan, 430074, China

[2]Key Laboratory of Ecosystem Network Observation and Modeling, Institute of Geographic Sciences and Natural Resources Research, CAS, Beijing, 100101, China

[3]State Key Laboratory of Biogeology and Environmental Geology, China University of Geosciences, Wuhan 430074, China;

[4]College of Resources and Environment, University of Chinese Academy of Sciences, Beijing 100049, China;

[5]Hubei Key Laboratory of Intelligent Geo-Information Processing, China University of Geosciences, Wuhan 430074, China

*Correspondence to: WANG Shaoqiang (sqwang@igsnrr.ac.cn)*

**Abstract.** Long-time series and large-scale crop calendars provide valuable information for rational crop cultivation and management under climate change scenarios as a prerequisite for ensuring regional food security. Although some Chinese rice phenological products exist, there is a lack of studies on the long rice cropping calendar for early, middle, and late rice according to the actual cropping dates. Unlike in-
situ monitoring and statistical rice phenology information, long-time series medium-resolution remote sensing satellite images provide the possibility and accuracy of real-time crop phenology monitoring. Based on MODIS data products and the improved PhenoRice algorithm, this study obtained phenological information on rice in seven major agricultural zones in China from 2003 to 2020. First, to effectively explore the changes in rice growing seasons over 18 years, we identified early, middle, and late rice
according to specific cropping dates. Second, we selected 30 % of the recorded phenology data from Agricultural Meteorological Stations (AMSs) for parameter calibration and added a season division module to the PhenoRice algorithm to obtain a 250 m resolution raster dataset of rice crop calendars named ChinaRiceCalendar. However, it is consistent with the station data, RiceAtlas, and ChinaCropPhen1km. The validation accuracy $R^2$ exceeded 0.95, 0.75, and 0.7 with the data recorded
from AMSs, RiceAtlas, and ChinaCropPhen1km, respectively. In addition, we observed that the rice planting dates in China were delayed by 2.4 days/10a while the maturity dates were earlier by 5.5 days/10a during 2003–2020. ChinaRiceCalendar provides insights into practical rice farming measures and the response of rice cropping dates to environmental conditions across China.

## Introduction

As one of the major food crops, rice feeds nearly half of the world's population, but only 12 % of the global cropland is covered by rice (Alexandratos and Bruinsma, 2012; Nelson and Gumma, 2015).





Therefore, information on rice cultivation is essential for various issues affecting human well-being, including food security, water use, and climate change in the Asian monsoon region (Nguyen, 2002; Rosenzweig and Parry, 1994). In the context of climate change, continued warming is projected to result in shorter growth periods and lower rice yields, leading to economic, ecological, and social problems (Asseng et al., 2011; Carleton, 2017; Penman et al., 2003). Under climate change scenarios, farmers' selection of appropriate planting times, suitable crop varieties, and suitable cropping patterns are essential in counteracting climate change risks (Waha et al., 2013).

A detailed crop calendar with specific records of crop varieties and key cropping dates is an integral part of the agricultural monitoring system and supports local farmers in planning proper crop cultivation (Fritz et al., 2019; Laborte et al., 2017; Mishra et al., 2021). Furthermore, exploring crop calendar changes can help optimize rice cultivation under climate change scenarios. In addition, access to large-scale crop calendars will contribute to the accuracy of crop modeling at regional and global scales(Bondeau et al., 2007; Stehfest et al., 2007).

Remote sensing, an effective tool for large-scale and long-time series crop phenology extraction combining crop growth characteristics with image features, is the basis for crop calendar establishment (Kotsuki and Tanaka, 2015; Mishra et al., 2021). In the process of phenological identification, phenological changes in the vegetation communities of the study image elements are obtained using satellite spectral vegetation indices (VIs) rather than specific vegetation characteristics (Luo et al., 2020; Piao et al., 2019; Xiao et al., 2006).

Curve-based and trend-based methods of phenological extraction depend on the changing status of the vegetation index(Gao and Zhang, 2021). The curve-based method to predict the current crop growth stage with historical crop phenology requires accumulating phenological information over a long period to obtain a robust standard change curve. A previous study used a two-step filtering method (TSF) to process MODIS broad dynamic range vegetation index data (WDRVI) standard curves of maize and soybean from 2003 to 2008, adjusting the parameters to fit the observed data and determine the crop growth period (Sakamoto et al., 2010). Furthermore, based on the data fusion of MODIS, Landsat, and Sentinel-2, the standard crop curve was obtained, and the cropping reference curve (CRC) method was used to obtain the crop distribution and phenological change status at 30 m (Sun et al., 2021). The trend-based method mainly relies on predetermined thresholds, curvature, and other information to judge the phenological stage, which does not depend on historical crop information and is flexible but less stable. The actual growth stage was obtained by analyzing the trend change of the curve through thresholds and curvature (Gao et al., 2020a; Liu et al., 2018). The trend-based method relies on pre-set thresholds, curvature, and other information to determine the weathering stage, which does not rely on historical crop information and is flexible but less stable. Another study used intra-seasonal emergence mapping (WISE) to determine the emergence date of crops using VENμS data and compared the differences of Sentinel-2 and HLS data (Gao et al., 2020a).

PhenoRice combines the advantages of both methods to monitor rice growth over a continuous period based on remote sensing phenology theory using medium-resolution images over a long period to analyze seasonal variation (Boschetti et al., 2017). Owing to its excellent performance, phenological information on rice has been extracted in rice-growing areas such as Nepal, India, Italy, and the Philippines, and it has high accuracy when compared with the RiceAtlas and other census-based rice cropping calendars(Busetto et al., 2019; Liu et al., 2020; Mishra et al., 2021). However, it should be noted that these studies explored the seasonal variation of rice based on the division of growing seasons only on artificially delineated rice heading dates, resulting in the possible duplication of the growing seasons of





rice, which needs to be reprocessed when compared with the actual rice crop (Mishra et al., 2021). Therefore, the regional application of PhenoRice first needs to delineate the actual growing seasons in conjunction with rice cropping practices to improve the accuracy of rice crop calendar extraction on a large scale.

Rice, a grain crop with China's largest sown area and highest total grain production (Peng et al., 2009). Rice cropping patterns in China are complex and diverse, including three rice-growing seasons (i.e., early, middle, and late rice) (Cao et al., 2021; He et al., 2021) and different cropping patterns (e.g., rice-oilseed rape, rice-tobacco) (Frolking et al., 2002; Qiu et al., 2003). However, we observed that the differences in rice types due to ignoring the actual cropping dates in previous studies made the current rice crop calendar

focus on the key phenological dates in mono seasons (Luo et al., 2020; Qiu et al., 2017) and did not explore the spatial and temporal distribution for different rice varieties in China. A high-resolution rice calendar dataset (ChinaCropPhen1km) already exists, and the classification of rice types in this dataset is based on cropping frequency and cropping dates (early, late, and one-season rice). However, the actual cropping dates of some one-season rice, including the other two varieties, make it challenging to explore

the spatial and temporal trends of different rice varieties. In addition, the 1 km pixel range is significantly affected by mixed pixels, which cannot reflect the regional rice cropping status. Meanwhile, current rice crop calendars in China are scarce. Owing to the limitations of historical agricultural census data, RiceAtlas is the most widely used global rice crop calendar but is limited to the province level in China, and further research is needed (Laborte et al., 2017). In recent decades, the demographic structure of

China's rural and urban areas has continued to change. The level of agricultural technology has continued to improve, resulting in corresponding adjustments in rice cropping patterns and practices (Cao et al., 2021; Wang et al., 2021; Zheng et al., 2020). The resulting changes in rice cropping patterns are critical to national food security, and exploring the changes in the cropping dates of rice has become the basis for rationalizing rice cultivation in China.

Therefore, to address the shortcomings of the existing rice crop calendar in China, we attempted to improve the PhenoRice algorithm and use satellite remote sensing data to (1) establish crop calendars for early, middle, and late rice in China; (2) validate the extracted rice sown areas and crop calendars in different growing seasons; and (3) explore the spatiotemporal changes of rice cropping dates in major agricultural zones from 2003 to 2020.

**Data and Methodology**

**Study area**

We selected seven agricultural zones in China as the study area: the Northeast Plain (DB), Huanghuaihai Plain (HH), Loess Plateau (HT), Middle and Lower Yangtze River Region (CJ), South China Region (HN), Yunnan-Guizhou Plateau (YG), and Sichuan Basin and Surrounding Region (SC) (Fig.1). The DB

zone planted with one season middle rice is susceptible to low-temperature stress. HH and HT zones are not major rice-producing zones. In the central and southern Huanghuaihai Plain, middle rice planting occurs. The zone has a significant diurnal temperature difference, little rainfall, and high evaporation. The CJ zone is located in the subtropics, and the rice varieties include one season middle rice and early-late rice. The HN zone has a high replanting frequency and generally grows two seasons of rice a year.

In comparison, the southern part of Hainan Province can grow three seasons of rice. Mountainous plateaus typically characterize the YG and SC zones. Rice is mainly grown in mountainous basins and



river valleys 800–2700 m.

## Data

### Satellite Imagery

MODIS data are widely used in crop phenology research because of their excellent performance in temporal and spatial continuity (Reed et al., 1994; Son et al., 2013; Zhang et al., 2003; Zhao et al., 2011). Although MODIS data have cloud cover in the tropics and subtropics, it cannot obtain cloud-free images of the study area for the entire period. The better temporal resolution makes MODIS the best choice for extracting crop calendars over long periods at a large scale. We selected two MODIS EVI products for

the study area during 2003–2020: MOD13Q1 and MYD13Q1 (https://doi.org/10.5067/MODIS/MOD13Q1.061, 250 m, 16-day). Because Terra and Aqua are based on the synthetic period of moving eight days from each other, the time series of the two 16-day products of MOD13Q1 and MYD13Q1 theoretically have a frequency of eight days (Boschetti et al., 2017). The red ($\rho_{RED}$) and near-red ($\rho_{SWIR}$) bands of MOD13Q1 and MYD13Q1 were selected for the Normalized

Flooding Index (NDFI) (Eq. 1). The pixel reliability quality indicators for data correction of the de-clouded blue (B3) band and the actual image obtained to date (DOY) were used. In addition, considering the physiological temperature constraints for rice growth, the 1 km spatial resolution and 8-day temporal resolution Land Surface Temperature (LST) product MOD11A2 (https://doi.org/10.5067/MODIS/MOD11A2.061) was selected by resampling 250 m spatial resolution

consistent with EVI data.

$$NDFI = \frac{\rho_{RED} - \rho_{SWIR}}{\rho_{RED} + \rho_{SWIR}} \qquad (1)$$

All data are based on the Google Earth Engine (GEE) platform to maximize computational efficiency and use the Python package of geemap for filtering and computation (Wu, 2020).

### Validation Data

First, we compared the rice sown areas extracted from the Chinese Agricultural Yearbook for early, middle, and late rice in each province from 2003 to 2020. The phenological records from Agricultural Meteorological Stations (AMSs, https://data.cma.cn/) were compared with the results of ChinaRiceCalendar by $R^2$ and RMSE to evaluate the accuracy of the estimated cropping dates at the station scale. This phenological information includes rice varieties (single and double seasons; early and

late rice) and critical phenological data (planting, flowering, and maturity dates) of more than 226 stations from 2003 to 2013.

In this study, we compared rice crop calendars with previous studies, including the RiceAtlas dataset obtained from the agricultural census (Laborte et al., 2017) and the ChinaCropPhen1km dataset extracted from satellite remote sensing data (Luo et al., 2020).

### Additional Data


Cropland data were selected from the International Geosphere-Biosphere Program (IGBP) classification of the MODIS land cover product (MCD12Q1, 500 m) from 2003 to 2020 (https://doi.org/10.5067/MODIS/MCD12Q1.006)to remove other land cover types.



Elevation data were obtained from the Shuttle Radar Topography Mission (SRTM, https://srtm.csi.cgiar.org) digital elevation model (DEM) data from the GEE platform, resampled to a spatial resolution of 250 m.

**Methodology**

The technology roadmap of this study is shown in Fig.2:

**Data preprocessing**

The data preprocessing process can be divided into four steps:

1. To effectively capture the flooding signal specific to rice before planting, this study emphasizes paddy rice, while dry-crop rice was not considered. Because wetlands and swamps may have similar flooding characteristics to rice fields (Dong and Xiao, 2016; Han et al., 2022), rice identification is affected. The annual cropland extent from 2003 to 2020 was used to establish a cropland mask to 170   crop the original MODIS EVI data.

2. Considering the environment, too high altitude and slope are unsuitable for average rice growth (Dong and Xiao, 2016; Gumma et al., 2011). Therefore, areas with an elevation below 2600 m and a slope less than 8° were screened as terrain masks (Han et al., 2022). Based on the data clipped by the cropland mask, the altitude and slope were screened to further narrow the study area to the 175   physiological and environmental conditions of rice growth

3. Due to cloud pollution in some MODIS images and severe cloud cover in low latitudes, the blue band (B3) was adopted for image cloud removal; the pixels with B3 ⩾ 0.2 were deleted (Xiao et al., 2006).

4. Because the actual EVI values are usually more significant than the contaminated pixels, to 180   effectively reduce the information of the contaminated pixels, the EVI time series reconstruction mainly relies on the information of the high-quality pixels. Therefore, the actual EVI values are classified into three categories: good, marginal, and bad, and their corresponding maximum ($W_{max}$), middle ($W_{mid}$), and minimum weight ($W_{min}$) values are 1.0, 0.5, and 0.2, respectively (Kong et al., 2022).

**Growing season division**

The PhenoRice algorithm must first incorporate the actual rice cropping information in the study area. Consequently, the rice phenology data recorded by 30% of AMSs were selected for parameter adjustment of the algorithm (Table.1). The coarse fitting method was used to capture seasonal rice signals and divide the approximate rice-growing season (Kong et al., 2022) to reduce the uncertainty caused by the artificial 190   division of growing seasons. The Whittaker smoothing method can stably capture seasonal vegetation signals with little noise interference, and it is widely used to identify crop phenology (Atzberger and Eilers, 2011; Bush et al., 2017). During the coarse fitting of the EVI time series, the primary reference was the wWHIT function in the phenofit (Kong et al., 2022).

**Extraction of rice growth information**

Rice is distinguished from other crops; it is usually transplanted or wet-sown in flooded fields for a few days and thrives for a specific period after flooding. However, in regions with complex crop patterns,





relying on this characteristic alone without considering crop varietal changes and actual local crop calendars and agricultural practices may overlook part of the growing season (Manfron et al., 2012; Peng et al., 2011). Therefore, we chose the PhenoRice algorithm, which combines the seasonal number with a

single seasonal phenological extraction. This operates in two steps: (1) First, it checks whether the pixel belongs to the rice pixel: the average value of the EVI series is below the predefined threshold ($EVI_{avg\_th}$). The maximum value is greater than the threshold ($EVI_{max\_th}$), and the minimum value is less than the threshold ($EVI_{min\_th}$) for at least three positive/negative first-order derivatives within a time window of five cycles from the end/beginning of the EVI maximum. (2) Determine this rice pixel's critical

phenological stage for a single growing season. The time difference between the EVI minimum and maximum within a single growing season obtained by coarse fitting is within the specified period ($\Delta t_{min}$, $\Delta t_{max}$), and the minimum temperature exceeds the threshold ($LST_{th}$). The flood signal was detected within a specified time window centered on the minimum (winfl). The EVI continued to increase after the minimum, and the decrease was greater than the threshold ($dec_{th}$) within the limited time window after

the maximum EVI ($win_{decr}$). Finally, the planting date of a single growing season within this pixel was the minimum closest to the retained maximum. The flowering date corresponded to the midpoint above the 90th percentile between the planting date and the EVI maximum. The maturity date was the first inflection point with a negative first-order derivative after the EVI maximum.

**Data validation**

This study classified rice into early, middle, and late rice based on the date of transplanting and maturity. Rice with a transplanting date of DOY30-130 and a growing period of 70–100 days was defined as early rice. The rice with a transplanting date of DOY110–180 and a growing period of 100–130 days was defined as middle rice. Finally, the rice with a transplanting date of DOY150-230 and a growing period of 130-150 days was defined as late rice.

We verified the extracted area from 2003 to 2020, the extracted rice area was classified and compared with the area of the provincial statistics. Extraction results for planting, flowering, and maturity dates of different rice varieties were searched within 1 km of the remaining 70 % of AMSs and compared with recorded data to verify the accuracy of the extraction results at the site scale. The ChinaCropPhen1km dataset was selected and compared with the results of this study by R2 at the pixel scale. Considering

data availability, the extracted results were compared with the RiceAtlas cropping date ranges for rice at the provincial scale. Finally, the spatial resolution of 250 m interannual raster rice (early, middle, and late) cropping date dataset was established.

To minimize the variability caused by the quality of remote sensing data, we analyzed the changes in cropping dates of different rice varieties in the rice crop calendar from 2003 to 2020 at the county scale.

**Result**

**Validation of rice sown area in different seasons**

Different rice had different extraction accuracy, with the $R^2$ of early and middle rice exceeding 0.9. The middle rice had the best accuracy of 0.963 compared to the annual rice sown area of different rice varieties in each province from 2003 to 2020 with the statistical data. In contrast, the result for late rice

was the worst (Fig.3). The accuracy of early and late rice in the CJ zone exceeded 0.95, but the $R^2$ of late rice was 0.527. The accuracy of the extracted middle rice in DB, HH, and HT exceeded 0.95. The





accuracy of early and late rice in HN was lower, 0.586 and 0.699, respectively, and the accuracy of early rice was lower than that of late rice. The results for early rice in SC and YG were better than those for late rice (Table.2).

**Validation of rice calendars**

In this study, rice phenology data from AMSs were divided into early, middle, and late rice. The rice cropping dates of ChinaRiceCalendar within 1 km of the site were compared. After comparing all three rice's key cropping dates, the accuracy exceeded 0.94, and the error days did not exceed 14 days (two weeks). Among them, the best results were obtained for maturity dates (Fig.4a). When the three rice

varieties were compared by phenological stage and the best results were obtained for middle rice, all exceeding 0.5, regardless of the stage. The accuracy of late rice (0.513) at the planting dates was stronger than that of early rice (0.474), but was not high overall. The maturity dates showed this trend, resulting in the maturity dates of late rice (0.336) being stronger than those of early rice (0.197). However, at the flowering dates, the accuracy of early rice was higher than that of late rice, at 0.596 and 0.312,

respectively.

The results of RMSE were within 14 days when comparing the different varieties of rice planting, flowering, and maturity dates recorded by the AMSs in the seven agricultural zones (Fig.5). All three rice varieties in the CJ zone were verified, especially the late rice with the maximum RMSE of maturity (10 days) and flowering dates (9.9 days). The DB zone was dominated by middle rice, and the RMSE

for the three dates was less than eight days, with the smallest (4.1 days) for the maturity date. The HH and HT zones were dominated by middle rice. The errors for the three cropping dates were within seven days, and both had the lowest (6, 5.4 days) errors for the flowering dates. The minimum was the maturity date of middle rice (4.3 days). The RMSEs of all three cropping dates of the three rice varieties in the YG zone were larger than eight days, except for the flowering dates of early rice (8.1 days) and the

maturity dates of late rice (10.9 days) and the RMSEs of all three rice varieties in the SC zone were higher by eight days than those of the other varieties and stages.

**Comparison with existing datasets**

To strengthen the reliability of the results, we selected rice cropping information from existing products to validate the result: first, comparing RiceAtlas' provincial rice cropping dates range information with

ChinaRiceCalendar showed a high degree of agreement between the two, but $R^2$ was lower than 0.7 at the early rice planting dates in the HN and SC zones, and the accuracy of the rest of the results was higher than 0.7. For planting dates, the results for middle rice were better overall (0.82) than those for late rice (0.71) and early rice (0.70). The maturity dates showed this trend: middle rice (0.83) was stronger than late rice and early rice (0.74 and 0.72). In contrast, the comparison results were worse overall in the HN

and SC zones for early rice-planting dates (0.63). Compared with the RiceAtlas of census nature, areas with larger differences appeared in the south of the Yangtze River, concentrated in Guangdong, Sichuan, and Guizhou provinces (Table.3). In addition, when compared with the rice phenology information extracted by ChinaCropPhen1km using LAI, the date error was within 30 days. The best extraction results were obtained for planting dates (0.721), followed by maturity dates (0.696), and a larger error of 0.579

for flowering dates (Fig. 6).

**Spatial distribution of rice calendar dates at the county scale**

Based on the spatial distribution of different rice varieties from 2003 to 2020, the distribution of early and late rice cultivation was relatively similar, concentrated in the CJ, SC, YG, and HN zones (Fig.7). Early rice was mainly planted in the southwestern part of the CJ zone, the western part of the SC zone, and the western part of the YG zone. In contrast, late rice was distributed in the northern part of the CJ and the central part of the YG zone. In contrast, middle rice is widely distributed in the study area and is grown in every agricultural zone, with DB, CJ, SC, and YG being the main agricultural zones. The most widely sown areas in the CJ zone are in the northeast and central parts of the zone, namely Jiangsu, Hubei, and Hunan provinces.

This study compares and analyses three cropping dates for various rice varieties at the county scale to avoid the problem of rice pixel identification caused by remote sensing images. The planting dates for early rice vary significantly from one zone to another, and northwestern Sichuan, Jiangxi, Jiangsu, and Guangdong provinces are later than DOY105 (Fig. 8a). In contrast, the planting dates of middle rice tend to advance from east to west, especially in Heilongjiang, Liaoning, Jiangsu, and Jiangxi provinces, where the planting dates are slightly later than DOY140 (Fig. 8b). In Jiangsu, Jiangxi, and Guangdong, late rice planting dates were later than in other zones, exceeding DOY160 (Fig. 8c).

In the flowering dates, early rice showed a trend of gradually delaying the flowering dates from the eastern coast to the inland, similar to the planting dates, and the flowering dates in northwestern Sichuan, central Yunnan, Guangxi, southern Guangdong, and Hainan provinces exceeded DOY185 (Fig.8d). For middle rice, the flowering dates in Guangdong, Guangxi, southern Shaanxi, Guizhou, and central Jiangsu provinces were earlier than in other zones. They did not exceed DOY185, and later in the DB, HH, and central CJ zones, rice flowering dates exceeded DOY200 (Fig.8e). Late rice flowering dates exceeded DOY230 in northwestern Sichuan, Chongqing, Yunnan, Hunan, Jiangxi, Anhui, and northern Jiangsu provinces and were earlier than DOY190 in central Jiangsu and western Hubei provinces (Fig.8f). The flowering dates were higher than those of DOY230 in Jiangxi, Anhui, and northern Jiangsu provinces.

The early rice in northwestern Sichuan, Chongqing, Yunnan, Anhui, and Jiangsu provinces had later maturity dates compared to those in other zones (Fig.8g). Middle rice matured earlier from the east to the west in the central regions (Fig.8h). Late rice showed prominent latitudinal characteristics, indicating that maturity dates were delayed from the northern to southern regions, in Yunnan, Guangxi, Guangdong, and Hainan provinces. Guangxi and the southern Guangdong and Hainan provinces matured later than DOY290 (Fig.8i).

For each agricultural zone, the differences in the three cropping dates of all rice were significant. However, for the DB, HH, and HT zones, all three cropping dates of middle rice were relatively stable (planting dates: DOY130 ± 21; flowering dates: DOY200 ± 14; maturity dates: DOY260 ± 14)(Fig.9). The overall planting dates in the YG zone were earlier than DOY200, with the average early rice planting at DOY60, middle rice at DOY125, and late rice at DOY150; the flowering dates were close (DOY180 ± 30). However, the maturity dates differed significantly, DOY180 for early rice, DOY250 for middle rice, and DOY290 for late rice. DOY110, 160, 170, middle, and late rice were planted close to each other; the flowering date were significantly different, from DOY160 for early rice to DOY220 for middle rice, and DOY250 for late rice; the maturity dates of all rice were close (DOY280 ± 21). The planting dates of early and late rice in the HN zone had a long interval, about DOY90 and DOY180, respectively; the average flowering dates were DOY180, DOY220, and DOY240. The maturity dates of early rice (DOY210) were separated from the maturity dates of middle and late rice. The latter two were close to



each other (approximately 10 days apart), and the average maturity dates are close to DOY260.

**Temporal changes in rice calendar date at the county scale**

To effectively investigate the temporal changes in cropping dates of rice in ChinaRiceCalendar from 2003 to 2020, this study analysed the time series trend changes of three cropping dates using a Sen+Mann-Kendall trend analysis at a significance level of 0.05. The three rice varieties showed an overall trend of delayed cropping dates for early rice, with significant regional differences in middle rice.
In contrast, the cropping dates of late rice showed an apparent advancement trend (Fig.10).

The planting dates of early rice showed an overall delayed tendency. However, there was a significant trend in Guangdong Province. The regional variation in middle rice planting was significant. Late rice planting occurred significantly earlier in Yunnan, Guizhou, and Guangxi provinces. The flowering dates of early rice showed an overall delayed trend of 18 years. The flowering dates of middle rice in Sichuan,
Yunnan, and northwestern Heilongjiang provinces were significantly earlier. All late rice showed earlier flowering dates except for northeastern Guangdong and southwestern Yunnan provinces. The maturity dates of early rice showed an overall trend of pushing back, with only the western part of Sichuan province showing advancement. The maturity dates of middle rice in Heilongjiang, Jiangsu, Anhui, and most counties in Yunnan and Guangxi provinces were advanced. The rest of the counties showed no
significant changes. The maturity dates of late rice showed advancement, especially in central Anhui and northern Jiangxi, and some counties in the central Guizhou provinces showed a significant advancement. All three rice groups are sown in the CJ zone, and all early rice cropping dates show a postponement trend. However, cropping dates for late rice advance significantly, and the number of counties with delayed and advancement of middle rice is close. The DB zone is planted with middle rice, the 18-year
cropping dates are delayed, and the maturity dates are advanced. Although only middle rice is sown in the HH zone, the trend of all cropping dates is insignificant. Over 60 % of the counties planted with rice have early flowering dates. The HN zone sown with rice differs significantly from the CJ zone's early rice planting dates and delayed flowering and maturity dates. The number of counties with delayed and advanced planting dates was similar for middle rice in the two zones, with more than 65 % having
advanced planting dates. However, the SC zone was delayed for the maturity dates of middle rice, and the YG zone was early (Fig.11).

**Discussion**

Early rice was mainly concentrated in the CJ, SC, HN, and YG zones. In contrast, late rice was distributed similarly to early rice. However, middle rice was grown in all agricultural zones, and it should be
emphasized that the typological rice varieties in this study, according to planting dates, maturity dates, and growth time, indicate that all three varieties were not grown in the exact location. This may be because the early and middle rice grows and develops, and the other vegetation starts to grow. The planting dates of late rice in this study are around DOY180; the information on other vegetation will interfere with the normal extraction of late rice, resulting in unsatisfactory late rice extraction (Luo et al.,
2020). In addition, late rice was concentrated south of the Yangtze River. The lack of remote sensing image data during the planting dates owing to cloudiness may be one of the reasons for the poor extraction effect (Clauss et al., 2016; Xiao et al., 2005). The extraction effect discussed for the agricultural zones supports this speculation, and the extraction accuracy of late rice was significantly lower than that of





early and middle rice. However, the extraction area of early rice in the HN zone was significantly lower
than that in other agricultural zones because rice in the HN zone is sown in hilly and mountainous areas,
such as Fujian and Guangdong provinces, with a fragmented distribution (Dong and Xiao, 2016; Xiao et
al., 2005).

In this study, the analysis of the extracted rice information and agrometeorological stations revealed that
all three key cropping dates were extracted with an accuracy of $R^2>0.9$, indicating the feasibility of
applying the improved PheoRice algorithm in China. However, comparing the data in different
agricultural zones, the RMSE of the HN zone exhibited an error of more than seven days. This was due
to its subtropical location, the large span of rice planting dates, and interannual fluctuations, such as the
early rice transplanting date of DOY15 at Lingshui station (Hainan). This might cover the end of the
previous year, which would generate a significant error when compared (Liu et al., 2020; Shi et al., 2013;
Xiao et al., 2005). In contrast, in the DB, HH, and HT zones, where rice is grown only in mid-season,
the station record data are stable and have better accuracy in comparison.

In addition to considering actual cropping dates to divide rice, ChinaRiceCalendar has further improved
spatial and temporal resolution accuracy and comparison with census data and agricultural site data.
Compared with RiceAtlas's rice cropping date range for provincial administrative units in China,
ChinaRiceCalendar records annual rice cropping dates at the pixel scale, and the trend changes of rice
cropping dates at the county scale from 2003 to 2020 can quantitatively describe the changes in the rice
cultivation period in China. Considering the mixed pixel, compared with the 1 km pixel size of
ChinaCropPhen1km, ChinaRiceCalendar increases the spatial resolution to 250m to avoid the influence
of other features in identifying rice cropping dates. In addition, when the data results of
ChinaCropPhen1km were compared with the site data using the same method, the overall accuracy of
ChinaRiceCalendar was significantly higher (0.80) than that of ChinaCropPhen1km (0.62). Therefore,
ChinaRiceCalendar is a more reliable current rice seasonal crop calendar product.

The rice planting dates were later, and the maturity dates were earlier, when comparing the trend of rice
cropping dates at the county scale from 2003 to 2020. Adjusting rice planting dates has become more
reliable for improving crop yield. Earlier rice planting can effectively reduce the adverse effects of high
temperatures on rice production (Kim et al., 2021). Delaying planting has effectively reduced the
negative effects of maximum and minimum temperatures in rice fields, increasing crop yield and
improving rice quality (Deng et al., 2015; Ding et al., 2020). However, the exact advance date depends
on the rice variety and sown area. Our study confirms that rice management in China is moving toward
earlier planting dates. Studies suggest that rice yields will decline dramatically if the crop planting dates
are not changed, which is vital for regional grain yield security (Zhang et al., 2016). However, it has been
suggested that early planting dates can lead to high-temperature stress, affecting rice seeds' normal filling,
yield, and quality (Ahmed et al., 2015). This trend is evidenced by studies on earlier rice maturity (Song
et al., 2010); for example, some researchers have observed that high temperatures lead to earlier maturity
and shorter growing seasons based on experimental (Tao et al., 2006) and modeling (Mahmood, 1997)
studies in China and Bangladesh. Global warming trends have shortened the growing season of rice in
China, resulting in earlier maturity dates for both early and late rice (Tao et al., 2006; Zhang et al., 2013).
Although anthropogenic management largely influences rice cultivation, climate change's impact on
various rice cultivations may be significant. For example, Tao found that the flowering and maturity
dates of early rice in China were regulated by temperature and precipitation during the reproductive
period (Tao et al., 2006). The variation in maturity date may be related to the reduction in solar radiation
during the growing season (Tao et al., 2013). In addition, it has been demonstrated that the nutritional



growth period of rice is positively correlated with photoperiod duration (Fukai, 1999; Vergara and Chang, 1985; Yin and Kropff, 1998). However, a related study found the strongest correlation between maturity dates, transplanting dates, and tasseling in Chinese rice, with a weaker effect of temperature (Zhang et al., 2016). Consequently, crop driver analysis and future scenario simulation under climate change scenarios are necessary based on the rice crop calendar results extracted from this study.

**Uncertainty analysis**

In this study, we used MODIS remote sensing images as the primary data source to extract and analyze rice cropping dates in agricultural zones in China. Although we have improved the accuracy and reduced the influence of other external factors, this study has some uncertainties and limitations. Such shortcomings may be caused by (1) the characteristics of MODIS remote sensing images; although remote sensing images have the advantage of real-time monitoring, the influence of cloud cover on image quality is unavoidable; (2) this study mainly uses the phenomenon of brief inundation at the time of rice transplanting for the preliminary identification of rice, and some dry rice cannot be effectively identified; and (3) although we improved the accuracy of rice cropping dates to 250 m, the different land cover types within a pixel will lead to inaccurate pixel identification. Therefore, high-precision satellite remote sensing data are a future development direction for large-scale rice cultivation condition extraction.

**Data Availability**

The 250 m resolution raster dataset of seasonal rice crop calendars can is available at https://doi.org/10.7910/DVN/EUP8EY (Hui Li, 2023). The spatial reference system of the dataset is EPSG:4326(WGS84).

**Conclusion**

As one of the world's most important agricultural countries, rice plays a crucial role in China. Due to the vast area of rice cultivation in China, there are significant regional differences in rice types and cropping patterns. In particular, in the context of the current international situation and climate change, rice cropping patterns in China have changed significantly. Therefore, a Chinese rice crop calendar with good spatial and temporal continuity will be helpful for rational planning and a varied selection of rice cultivation. However, a reasonable extraction method and rice variety data have not been applied on a large scale in China. Therefore, we improved the rice-growing season division module of the PhenoRice algorithm for extracting the rice calendar by cropping dates on a large scale in China at present and completed ChinaRiceCalendar from 2003 to 2020. The validation accuracy R2 exceeded 0.95, 0.75, and 0.7 with the data recorded from AMSs, RiceAtlas, and ChinaCropPhen1km, respectively. The validation results proved the advantages of the extracted rice crop calendar in the Chinese context. In addition, we compared the variation of rice cropping dates for 18 years and demonstrated that the rice planting date was significantly delayed (2.4 days/10a) and the trend of maturity date advance was significant (5.5 days/10a). The generated rice crop calendar will provide data support for rice management and related industrial development, such as yield and income increases under future climate scenarios in China.



**Author Contributions:** Conceptualization, methodological and algorithmic improvements, L.H. and W.X.; data download and processing, L.Y. and L.Z.; validation, C.S. and W.Q.; formal analysis, Z.T. and L.H.; writing-original draft preparation, L.H. and W.X.; writing-review and editing, W.S., W.L. and W.L.; supervision project administration, W.S. All authors have read and agreed to the published version of the manuscript.

**Funding:** This research was funded by the National Natural Science Foundation of China (Project No. 31861143015).

**Acknowledgments:** The authors would like to thank NASA for providing the MODIS data.

**Conflicts of Interest:** The authors declare no conflict of interest.

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



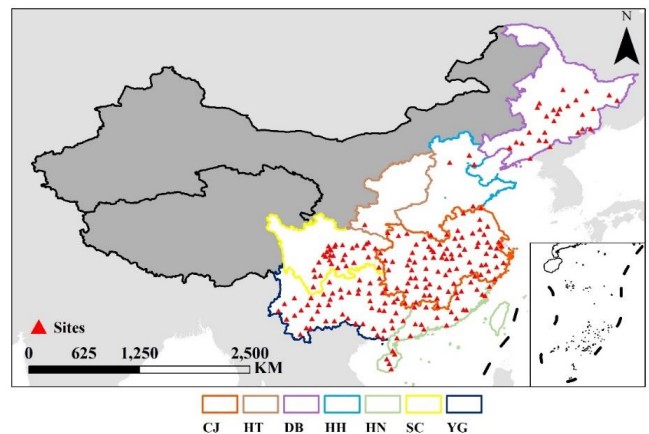

**Fig.1 Study area and distribution of Agricultural Meteorological Stations (AMSs) in China**

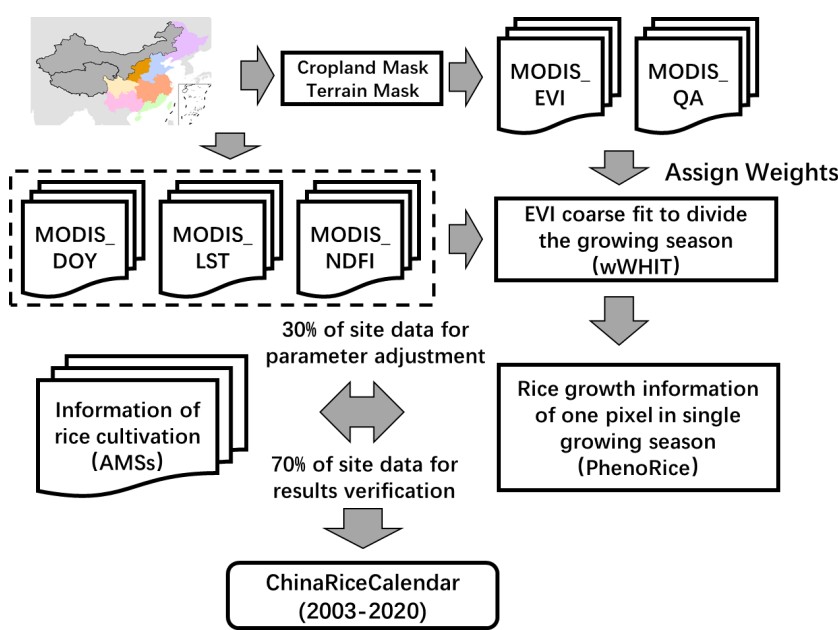

605                                  **Fig.2 Technology roadmap for this study**

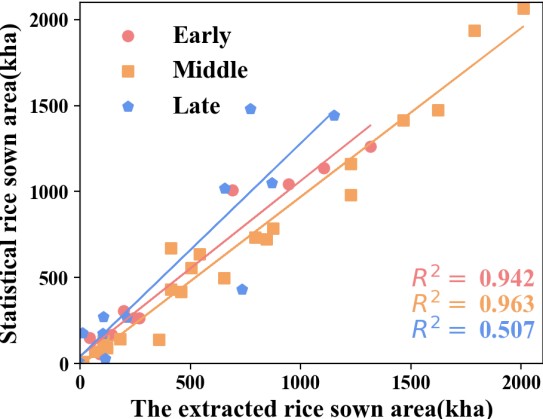

**Fig.3 Comparisons of extracted rice sown area with the statistical sown area at the province scale (red dots for early rice, orange squares for middle rice, blue pentagons for late rice, p < 0.05)**

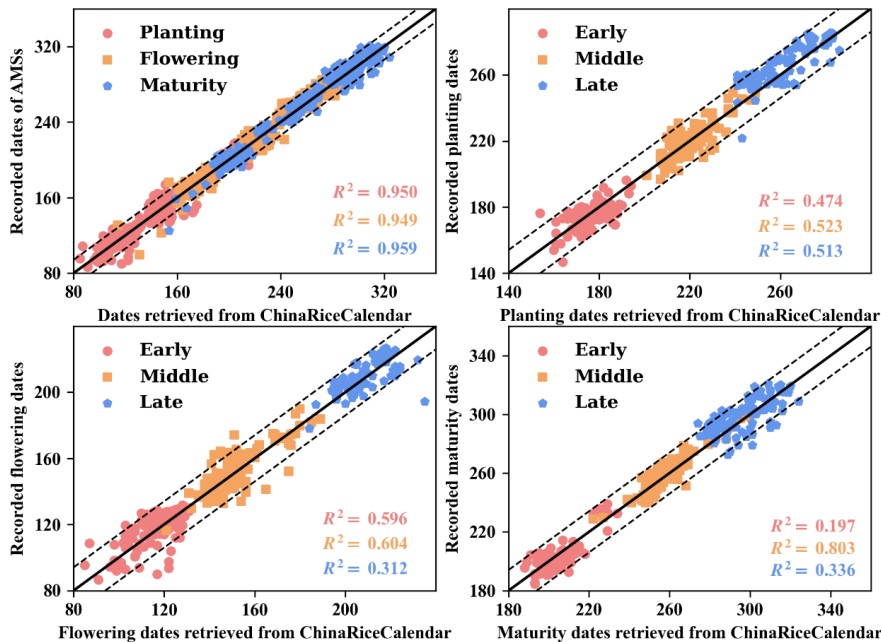

**Fig.4 Comparison of rice cropping dates of ChinaRiceCalendar with AMSs recorded data at the site scale (dashed lines are ±14 days, p < 0.05)**

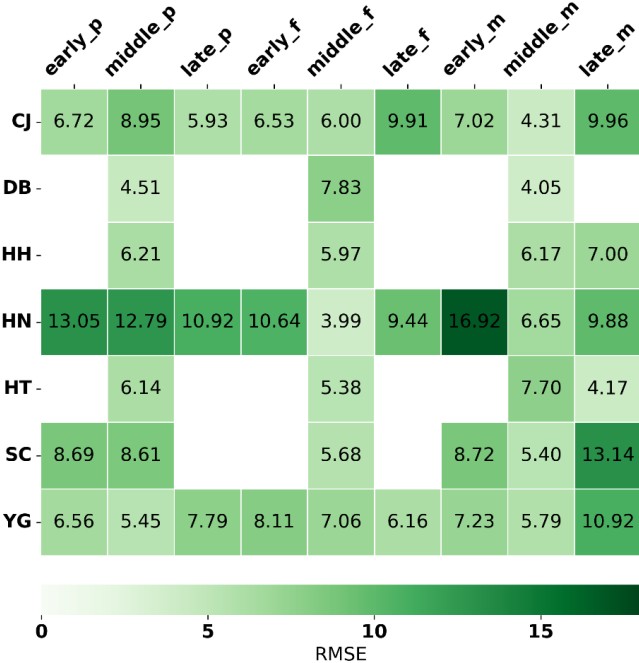

**Fig.5 RMSE of rice cropping dates from ChinaRiceCalendar in main agricultural zones with recorded data of AMSs**

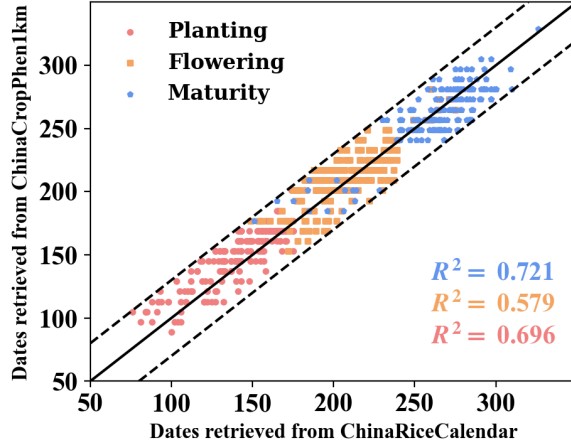


**Fig.6 Comparison of ChinaRiceCalendar and ChinaCropPhen1km at the pixel scale (dashed line: ± 30 days, p<0.05)**



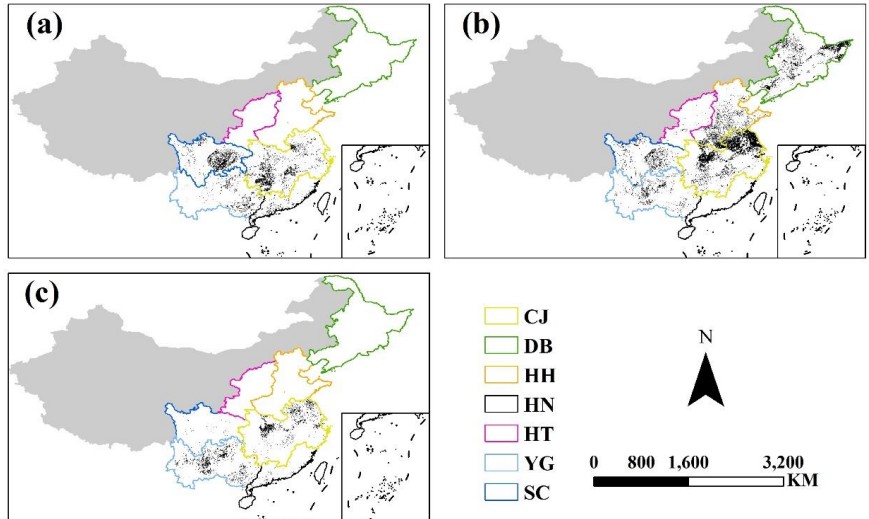

**Fig.7 Spatial distribution of different varieties of rice (a: early rice, b: middle rice, c: late rice)**

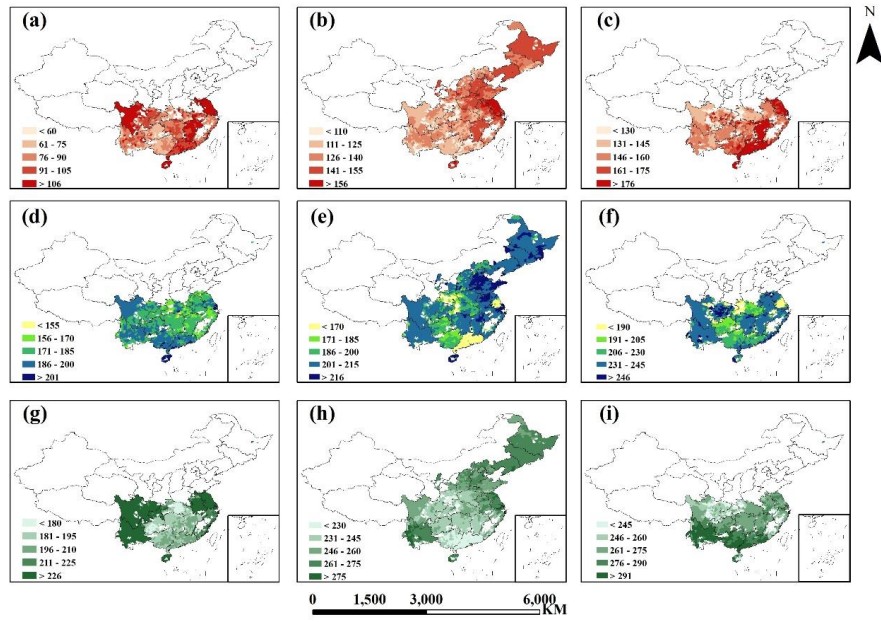


**Fig.8 Rice cropping dates at the county scale from 2003 to 2020 (a: early rice planting dates; b: middle rice planting dates; c: late rice planting dates; d: early rice flowering dates; e: middle rice flowering dates; f late rice flowering dates; g: early rice maturity dates; h: middle rice maturity dates; i: late rice maturity dates)**

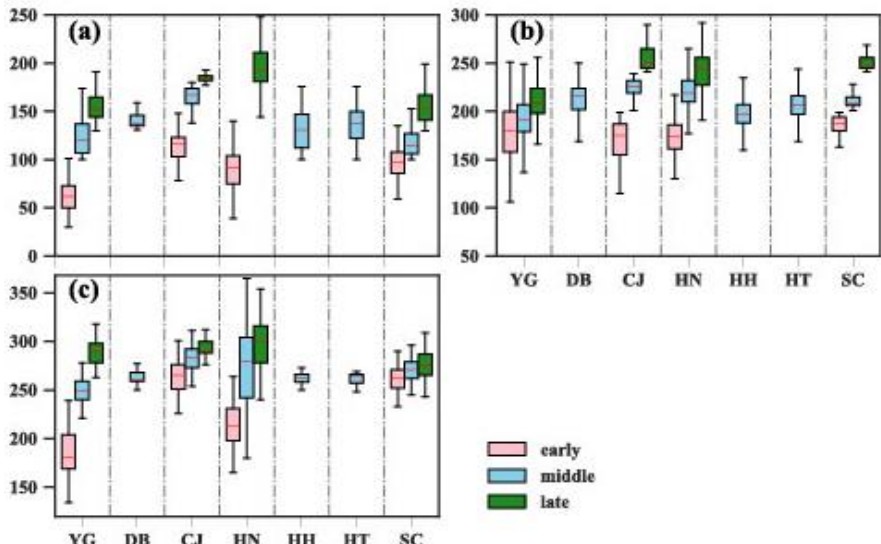

**Fig.9 Rice cropping dates in main agricultural zones at the county scale from 2003 to 2020 (a: Planting; b: Flowering; c: Maturity)**

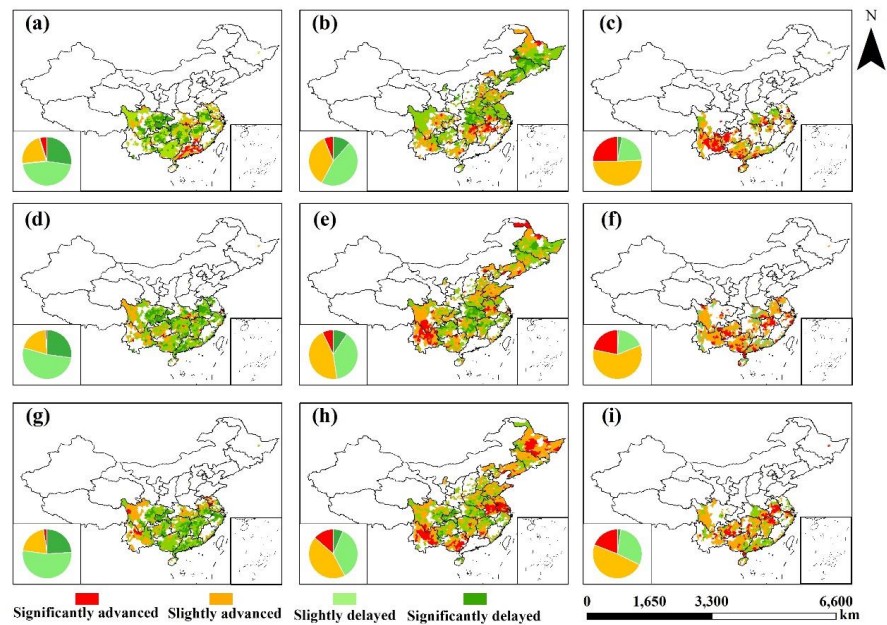

**Fig.10 Temporal changes in rice calendar date at the county scale from 2003 to 2020 ((a: early rice planting dates; b: middle rice planting dates; c: late rice planting dates; d: early rice flowering dates; e: middle rice flowering dates; f late rice flowering dates; g: early rice maturity dates; h: middle rice maturity dates; i: late rice maturity dates)**

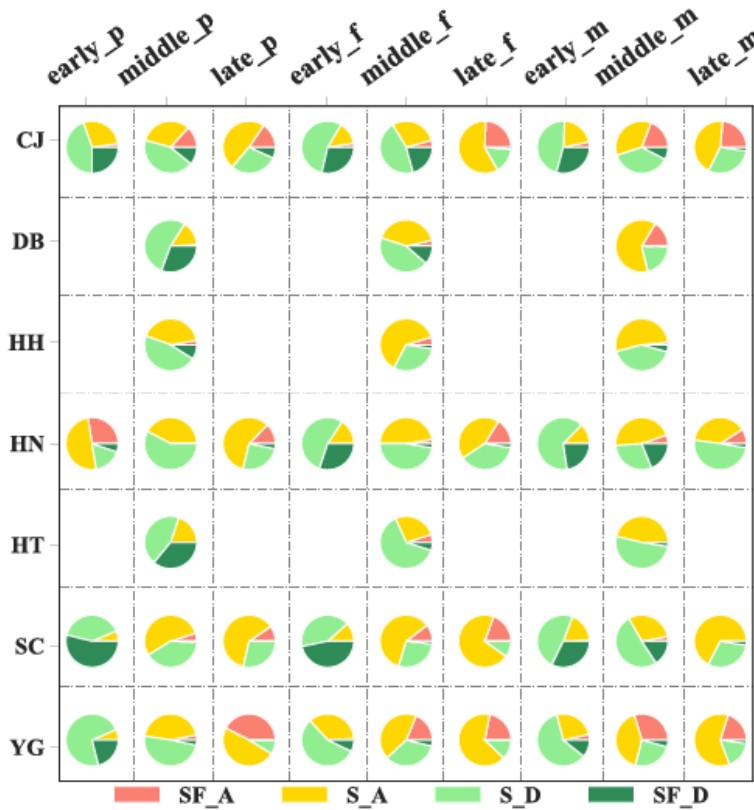

**Fig.11 Temporal changes of rice cropping dates in main agricultural zones at the county scale from 2003 to 2020 (SF_A: Significant advanced; S_A: Slightly advanced; S_D: Slightly delayed; SF_D: Significantly delayed)**




**Table.1 Description of the parameters in the PhenoRice algorithm**

| Parameters | Value | Description |
|---|---|---|
| $EVI_{avg\_th}$ | 0.4 | The average EVI value during the study period needs to be greater than the threshold |
| $EVI_{max\_th}$ | 0.5 | The maximum value of EVI during the study time period needs to be greater than the threshold |
| $EVI_{min\_th}$ | 0.25 | The minimum value of EVI during the study time period needs to be less than the threshold value |
| $\Delta t_{min}(8days)$ | 13 | The shortest time between seeding and peak |
| $\Delta t_{max}(8days)$ | 23 | The longest time between seeding and peak |
| $LST_{th}$ | 15 | Minimum land surface temperature for rice growth |
| $winfl(8days)$ | 3 | Window threshold for capturing flood signals |
| $minndfi$ | 0 | Minimum NDFI threshold |
| $win_{decr}(8days)$ | 9 | Threshold of drop window after maximum value |
| $dec_{th}$ | 0.55 | Decrease amplitude threshold within the specified time window after the EVI maximum value |

**Table.2 $R^2$ of the extraction rice sown area in main agricultural zones ($p < 0.05$)**

|  | Early Rice | Middle Rice | Late Rice |
|---|---|---|---|
| CJ | 0.956 | 0.963 | 0.527 |
| DB |  | 0.957 |  |
| HH |  | 0.989 |  |
| HN | 0.586 |  | 0.699 |
| HT |  | 0.981 |  |
| SC | 0.912 | 0.934 | 0.638 |
| YG | 0.824 | 0.879 | 0.435 |


**Table.3 Comparison of ChinaRiceCalendar with RiceAtlas $R^2$ in main agricultural zones**

|  | Planting | | | Maturity | | |
|---|---|---|---|---|---|---|
|  | Early Rice | Middle Rice | Late Rice | Early Rice | Middle Rice | Late Rice |
| CJ | 0.77 | 0.79 | 0.70 | 0.73 | 0.82 | 0.67 |
| DB |  | 0.957 |  |  | 0.94 |  |
| HH |  | 0.85 |  |  | 0.79 |  |
| HN | 0.68 | 0.72 | 0.52 | 0.70 | 0.81 | 0.64 |
| HT |  | 0.87 |  |  | 0.79 |  |
| SC | 0.63 | 0.79 | 0.81 | 0.81 | 0.80 | 0.77 |
| YG | 0.73 | 0.78 | 0.80 | 0.71 | 0.83 | 0.82 |