# Peer review of "ChinaRiceCalendar-Seasonal Crop Calendars for Early,"

_Earth System Science Data, 2023_

## Author Comment (AC2)

Dear Editors and Reviewers:

Thank you again for your comments on our work. We have revised our manuscript according to the advice from Referee #3. The responses are shown as follows:

Referee #3: The manuscript has shown notable improvements in clarity and organization. Regarding the methodology, the division of China's rice growing season into early, middle, and late phases, though new, might be readily adaptable by other algorithms. Therefore, the challenge lies in substantiating how this approach enhances existing algorithms. While the authors have added the comparison of the proposed algorithm's performance with existing ones, it would benefit from more detailed elaboration on the specifics of this comparison, specifically in how the comparisons were conducted and the results obtained. Inclusion of additional figures, akin to Figure 4, and a consolidated figure showcasing all algorithm results, would be encouraged. Additionally, it is crucial to employ a consistent resolution when generating the crop calendar data using the other algorithms for a fair comparison. A clearer understanding of the paper/data's contribution to the field could be achieved with a comprehensive presentation of the comparison results. Consequently, I recommend another major revision to facilitate a more informed decision-making process.

Thank you very much for the comments. According to your suggestions, we employed a consistent data resolution, validation approach, and benchmark for different rice calendars to reveal advantages of our ChinaRiceCalendar dataset. We compared the accuracy of multi-season calendar datasets on annual and seasonal scales and added Figure 5 into the manuscript. In China, our calendar dataset demonstrates high accuracy across all three rice seasons, while ChinaCropPhen1km exhibits suboptimal performance in early-rice seasons, RiceAtlas underperforms in middle-rice seasons, and RICA falls short in both middle- and late-rice seasons. Actually, our estimation yields superior results not solely due to the categorization of early, middle, and late seasons in China, but also stems from the localized algorithm parameters based on the phenological characteristics of early, middle, and late rice in each province (Table 1). Comprehensively, the pre-identification of potential growing periods, the localization of PhenoRice parameters, and the segmentation of rice seasons contribute to good performance of ChinaRiceCalendar in early, middle, and late rice.

[revised manuscript text omitted]

---

## Author Response (AR1)

✧  Blue text: Comments from reviewers
  ✧  Black text: Responses from authors
  ✧  Red text: Revised paragraphs in the new manuscript

Dear Editors and Reviewers:

We appreciate all of the valuable comments from the reviewers of our work. We have revised our manuscript according to your advice, and invited an expert to thoroughly polish the English language of the manuscript. The point-to-point responses to the reviewers' comments, questions, and suggestions are shown as follows:

**Reviewer #1:**

This study presents an interesting topic to generate a rice calendar dataset that depicts planting, flowering, and maturity in China from 2003 to 2020. While this study has some merits, some places need improvement.

1. In the abstract, it said that "there is a lack… cropping dates." However, there are several studies addressing this issue. For example: Liu, Y., Zhou, W. & Ge, Q. Spatiotemporal changes of rice phenology in China under climate change from 1981 to 2010. Climatic Change 157, 261–277 (2019). https://doi.org/10.1007/s10584-019-02548-w; Bai, H., Xiao, D. Spatiotemporal changes of rice phenology in China during 1981–2010. Theor Appl Climatol 140, 1483–1494 (2020). https://doi.org/10.1007/s00704-020-03182-8; Li S H, Xiao J T, Ni P, Zhang J, Wang H S, Wang J X. Monitoring paddy rice phenology using time series MODIS data over Jiangxi Province, China. Int J Agric & Biol Eng, 2014; 7(6): 28 −  Therefore, the contributions of this study need to be reconstructed.

Thank you very much for the comment. we rewrote the shortcomings of previous rice calendar studies, clarified the significance of accurately dividing rice growing seasons, and revealed the advantages of ChinaRiceCalendar over RiceAtlas, ChinaCropPhen1km, and RICA in the new manuscript.

Firstly, we modified the abstract (Lines 17-30):

**Abstract.** Long-time series and large-scale rice calendar datasets provide valuable information for agricultural planning and field management in rice-based cropping systems. However, current regional-level rice calendar datasets do not accurately distinguish between rice seasons in China, causing uncertainty in crop model simulation and climate change impact analysis. based on satellite remote sensing and an improved PhenoRice algorithm, we extracted the crop areas and phenology of early-, middle-, and late-season rice across China from 2003 to 2020, and established a multi-season rice calendar dataset named ChinaRiceCalendar. Overall, the ChinaRiceCalendar dataset shows a good agreement not only with field-observed rice calendars in Agricultural Meteorological Stations (AMSs), but also with statistical rice areas in various growing seasons. According to the calendar data from 2003 to 2020, the transplanting dates for

early, middle, and late rice shifted by +5.4, +2.6, and -5.7 DOY/decade, respectively; the flowering date for early, middle, and late rice shifted by +5.5, -2.8, and -2.7 DOY/decade, respectively; the maturity date for early, middle, and late rice shifted by +3.2, -3.6, and -5.1 DOY/decade, respectively. The ChinaRiceCalendar can be utilized to investigate and optimize the spatio-temporal structure of rice cultivation in China under climate and land-use change.

Secondly, we reconstructed the Introduction section and added relevant references into the third paragraph of the Introduction section. We stated the shortcomings of previous rice calendar studies in the Introduction section as follows (Lines 58-85):

'The PhenoRice algorithm excels at extracting rice phenology in multiple cropping systems and has been widely used in East Asia, South Asia, Southeast Asia, and Europe (Busetto et al., 2019; Liu et al., 2020; Mishra et al., 2021). However, the performance of the PhenoRice algorithm depends on the division of rice seasons, which requires expert knowledge about rice-based cropping systems in different regions (Mishra et al., 2021).'

'In China, there are at least three rice-growing seasons (early, middle, and late seasons) in diverse rice-based cropping systems (e.g., single-rice, double-rice, rice-wheat, rice-rapeseed, and rice-vegetable systems) (Frolking et al., 2002; Qiu et al., 2003; Cao et al., 2021; He et al., 2021). Generally, early-, middle-, and late-season rice are transplanted at Day Of Year (DOY) 30-130, DOY 110–180, and DOY 150-230, respectively. The growth periods after transplanting for early, middle, and late rice are 70-100 days, 100-130 days, and more than 130 days, respectively. Although field observations are important data sources for studying rice calendars in different growing seasons, they are usually limited by spatial and temporal discontinuities (Zhao et al., 2016; Wang et al., 2017). Therefore, previous studies have typically utilized satellite remote sensing products to establish rice calendar datasets at the regional scale (Shihua et al., 2014; Liu et al., 2019; Bai and Xiao, 2020; Luo et al., 2020; Mishra et al., 2021). Nevertheless, these calendar datasets based on satellite remote sensing do not rationally classify rice growing seasons across China. For example, the dataset ChinaCropPhen1km only distinguishes between early and late rice in double-rice systems (Luo et al., 2020); the assumptions of the dataset RICA about rice flowering dates in different seasons do not correspond to the realities in China (Mishra et al., 2021); Shen et al. (2023) produced high-resolution distribution maps of single-season rice but did not explore multiple rice cropping systems. Early-, middle- and late-season rice in China are not only planted at different times, but also have distinguishing varietal characteristics, such as different temperature and photoperiod sensitivities (Zong et al., 2021). Thus, a crop calendar that accurately classifies rice seasons will provide reliable data for agricultural models to calibrate crop parameters at the variety level. Moreover, effective identification of different rice seasons will help analyze the response and adaptation of rice phenology to climate change.'

Thirdly, we revealed the advantages of our ChinaRiceCalendar database as follows (Lines 260-278):

'**3.3   Comparison with other calendar datasets**'

'The rice phenological dates obtained from the RiceAtlas (Laborte et al., 2017),

ChinaCropPhen1km (Luo et al., 2020), and RICA (Mishra et al., 2021) datasets were also validated against the spatially corresponding AMS data. In China, the rice phenological dates estimated in ChinaRiceCalendar show higher accuracy by growing season than those obtained from RiceAtlas and RICA. Furthermore, compared to the ChinaCropPhen1km dataset, the ChinaRiceCalendar dataset have similar accuracy in rice phenological dates but higher accuracy in rice areas in different growing seasons.'

'The RMSE between RiceAtlas' and AMSs' phenological dates for early, middle, and late rice in China is 18.27, 21.03, and 13.81 days, respectively. The $R^2$ between RiceAtlas' and AMSs' phenological dates for early, middle, and late rice in China is 0.65, 0.43, and 0.75, respectively.'

'The RMSE between ChinaCropPhen1km's and AMSs' phenological dates is 9.35 days for early and middle rice, and 7.24 days for late rice in China. The $R^2$ between ChinaCropPhen1km's and AMSs' phenological dates is 0.81 for early and middle rice, and 0.85 for late rice in China.'

'The RMSE between RICA's and AMSs' phenological dates for early, middle, and late rice in China is 22.80, 14.07, and 13.61 days, respectively. The $R^2$ between RICA's and AMSs' phenological dates for early, middle, and late rice in China is 0.47, 0.69, and 0.73, respectively.'

2. The introduction did not well review previous research. Several classic phenological extraction methods are missing. Such as: Jönsson, P. and Eklundh, L., 2004, TIMESAT - a program for analysing time-series of satellite sensor data, Computers and Geosciences, 30, 833-845; Zhang, X., M. A. Friedl, C. B. Schaaf, A. H. Strahler, J. C. F. Hodges, F. Gao, B. C. Reed, and A. Huete (2003), Monitoring vegetation phenology using MODIS, Remote Sens. Environ., 84, 471–475.

3. The advantages and disadvantages of curve-based and trend-based methods are not clear, therefore the benefits of PhenoRice are uncertain.

Thank you for the two comments above. We rewrote the review regarding crop phenology detection methods and added the relevant literature (Lines 42-62):

'Satellite remote sensing is an effective tool for detecting long-term trends in crop phenology at the regional scale (Xiao et al., 2006; Kotsuki and Tanaka, 2015; Luo et al., 2020; Gao and Zhang, 2021; Mishra et al., 2021). Crop phenology detection methods based on remote sensing vegetation indices (VIs) can be categorized into threshold, inflection point, and shape model approaches. The threshold approaches assume that a development stage begins when the VI value exceeds a predefined threshold (Jönsson et al., 2004; Boschetti et al., 2009; Pan et al., 2015; Guo et al., 2016). The inflection point approaches reconstruct the VI time-series curve by filter smoothing or function fitting, and then corresponds the maxima, minima, and inflection points on the curve to the key phenological events (Zhang et al., 2003; Sakamoto et al., 2005; Sun et al., 2009; Wang et al., 2019). The shape model approaches fit observed VI time-series curves by geometric scaling a robust standard VI time-series curve for the specific crop to identify development stages (Sakamoto et al., 2010; More et al., 2016; Zeng et al., 2016; Sakamoto et al., 2018). In addition to the methods based on time series of VIs, there are also rule-based algorithms

that integrate multiple approaches and indicators to detect crop phenology, such as the PhenoRice algorithm proposed by Boschetti et al. (2017). The PhenoRice algorithm, which combines the advantages of threshold and inflection point approaches, utilizes the Enhanced Vegetation Index (EVI), the Normalized Difference Flood Index (NDFI), and the land surface temperature (LST) to estimate rice planting dates. The PhenoRice algorithm excels at extracting rice phenology in multiple cropping systems and has been widely used in East Asia, South Asia, Southeast Asia, and Europe (Busetto et al., 2019; Liu et al., 2020; Mishra et al., 2021). However, the performance of the PhenoRice algorithm depends on the division of rice seasons, which requires expert knowledge about rice-based cropping systems in different regions (Mishra et al., 2021).'

4. Please define the early, middle, and later rice here, as they are the first time mentioned.

'Early-, middle-, and late-season rice are transplanted at Day Of Year (DOY) 30-130, DOY 110–180, and DOY 150-230, respectively. The growth periods after transplanting for early, middle, and late rice are 70-100 days, 100-130 days, and more than 130 days, respectively.' (Lines 67-69)

5. In the Study area section, the abbreviations for the seven zones are not appropriate. They are not related to the full name. Rather than using DB for Northeast Plain, it may be better to use NP. Meanwhile, for readers who are not familiar with China, it will be better to put all the place names that are mentioned in the text on a map.

The abbreviations of the study zones were revised according to your advice and the place names were indicated on Figure 1.

[Figure]

**Fig.1 Study area and distribution of Agricultural Meteorological Stations (AMSs) in China**

6. If MODIS cannot provide cloud-free images of the study area for the entire period, why did authors use it?

This is an error in English presentation and we have corrected the relevant part of the MODIS data description.

7.  Please clarify DOY, is it the day of year?

Yes, it is. A description of DOY was added on Line 67.

8.  The purpose of using elevation data needs to be mentioned.

We stated the purpose of using elevation data on Lines 145-147:

'Given that too high an elevation or too great a slope is unsuitable for paddy rice cultivation (Gumma et al., 2011; Dong and Xiao, 2016), only the image pixels with an elevation below 2600 m and a slope less than 8° were selected to extract rice calendars (Han et al., 2022).'

9. To determine the rice pixel, I am not sure if the threshold method is a good option especially since EVI values may be influenced by several factors in different years, such as climate. Why not consider using some advanced method, such as the one mentioned here: Shen, R., Pan, B., Peng, Q., Dong, J., Chen, X., Zhang, X., Ye, T., Huang, J., and Yuan, W.: High-resolution distribution maps of single-season rice in China from 2017 to 2022, Earth Syst. Sci. Data, 15, 3203 – 3222, https://doi.org/10.5194/essd-15-3203-2023, 2023. This method may improve the accuracy of late rice.

Thank you for your advice. The PhnoRice algorithm is a rule-based algorithm that integrates multiple approaches and indicators to detect crop phenology. In the first draft of the manuscript, the description of the PhenoRice algorithm was incomplete. So we modified the description of the PhenoRice algorithm in the Introduction section as follows (Lines 53-62):

'In addition to the methods based on time series of VIs, there are also rule-based algorithms that integrate multiple approaches and indicators to detect crop phenology, such as the PhenoRice algorithm proposed by Boschetti et al. (2017). The PhenoRice algorithm, which combines the advantages of threshold and inflection point approaches, utilizes the Enhanced Vegetation Index (EVI), the Normalized Difference Flood Index (NDFI), and the land surface temperature (LST) to estimate rice planting dates. The PhenoRice algorithm excels at extracting rice phenology in multiple cropping systems and has been widely used in East Asia, South Asia, Southeast Asia, and Europe (Busetto et al., 2019; Liu et al., 2020; Mishra et al., 2021). However, the performance of the PhenoRice algorithm depends on the division of rice seasons, which requires expert knowledge about rice-based cropping systems in different regions (Mishra et al., 2021).'

Furthermore, we detailed the PhenoRice algorithm and our improvements to it (Lines 151-205):

**'2.3.2  Estimation of rice area and cropping calendar'**

[revised manuscript text omitted]

Also, we added the literature published by Shen et al. (2023) on Lines 78-79.

10. There are many grammar issues. For example, line 48: it needs a space between 'scales' and '(Bondeau et al., 2007…'; Line 78-70: the sentence is duplicated; Line 85: there is no verb; Line 121, what is 800-2700 m. Please carefully polish the article.

Thank you for your reminder. We have invited an expert to thoroughly polish the English language of the manuscript.

**Reviewer #2:**

This manuscript introduces a new dataset for seasonal crop calendar for major agricultrual regions in China. The paper is easy to follow. The methods and results are well presented. The data are accessible. The authors may address the following issues to further improve the paper:

1. The map in Figure 1 needs to be improved. Please add a description to note source of the agricultural region boundary data. I suggest using the solid color for each region and adding a province-level boundary layer on the map.

Thank you for your advice. We have modified Figure 1 according to your advice. Also, 'Agricultural zoning data were obtained from Resources and Environment Science and Data Center (https://www.resdc.cn/data.aspx?DATAID=275).' (Lines 100-102)

[Figure]

**Fig.1 Study area and distribution of Agricultural Meteorological Stations (AMSs) in China**

2. The flowchart illustrated in Figure 2 is kind of confusing for readers. What does the input (the map) mean? What is the meaning of different shapes for each process? Is the output data (2003-2020) produced recursively or at one time? What does the two-way arrow mean?

Thank you for your questions. The output data was generated at one time. We redrew the flowchart to make the process clearer:

[Figure]

**Fig.2 Technology roadmap for this study**

3. Please use a high-resolution image for Figure 9.

Thank you for your reminder. The Figure has been replaced with a higher-resolution image (Figure 8 in the new manuscript).

4. The discussion section is quite long and hard to follow. Please divide the discussions into several subsections (e.g., advantages, uncertainty, limitations, and future works) to help readers follow the content more smoothly.

Thank you for the comment. We have streamlined the Discussion section to emphasize the advantages and uncertainties of our dataset. We reorganized the Discussion section as follows (Lines 322-371):

'Although the generated dataset ChinaRiceCalendar shows an advantage in rice season identification, there is still uncertainty in the data source and phenology detection methods.

This study used MODIS remote sensing data to extract rice phenological dates in various growing seasons in China. The MODIS remote sensing products have an appropriate temporal resolution, long time series, and good time consistency for analyzing changes in rice calendars at the regional scale. Moreover, the MODIS data are easy to obtain and process on the GEE platform, allowing for automated and timely updating of the calendar dataset. However, the pixel-based detection of rice areas may be interfered with by the contamination of clouds, aerosols, and water vapor, especially during the monsoon season when rice is by far the dominant crop (Xiao et al., 2014; Mishra et al., 2021). Because synthetic aperture radar (SAR) can penetrate through clouds,

subsequent studies could combine optical and SAR images to avoid the impacts of clouds (Shen et al., 2023). Also, most paddies in southern China are smaller than the spatial resolution of MODIS data, which may result in a rough estimation of rice areas. Generating more satellite remote sensing products with higher spatial resolution and integrating multiple data sources from satellite-airborne-ground observations will facilitate real-time monitoring of rice cropping areas at the regional scale (Zheng et al., 2022; Sun et al., 2023). Additionally, precisely corresponding the image pixels from the MODIS dataset to the Agricultural Meteorological Stations remains a challenge during data validation. In the future, it would be beneficial to conduct a quantitative assessment to determine the representativeness of the MODIS pixels surrounding the AMS site.

In this study, we improved the method of growing season division in the PhenoRice algorithm. We also attempted to remove non-paddy pixels and reduce the impacts of low-quality data on the reconstruction of EVI time-series curves. Nevertheless, since the PhenoRice algorithm detects rice pixels by agronomic flooding signals, rainfed or upland rice systems will be much harder to detect. In China, rice is mainly planted in flooded paddy fields (Luo et al., 2022), which mitigates the problems of detecting rainfed or upland rice. Moreover, the VI-curve smoothing methods perform differently in different regions (Luo et al., 2020). To enhance the identification of rice growing seasons in multi-cropping areas, we suggested identifying the optimal smoothing method for MODIS-EVI time series in various rice-based cropping systems. Although the local tuning of the PhenoRice algorithm parameters could further improve the results, we employed a single configuration of temporal windows and threshold values across China because automated methods that perform robustly are essential for developing timely information about crop calendars over large extents (Mishra et al., 2021).

The uncertainty in crop area estimation is more significant for late rice than for early and middle rice, resulting in lower accuracy of the detected rice area in southern China (MLY, SC, SCS, YGP) than in northern China (NP, HP, LP). For example, there is an underestimation of the double-season late rice area in Hainan Province and an overestimation of the single-season late rice area in Hubei Province. Because the transplanting dates (DOY150-210) of late rice coincide with the rainy season in the main rice-producing areas, there is a higher risk of misidentifying agronomic flooding signals during transplanting for late rice than for early and middle rice. Furthermore, low data quality induced by cloud contamination during the transplanting period contributes to the difficulties in extracting the late rice area (Xiao et al., 2005; Clauss et al., 2016). Also, the diverse multi-cropping systems and the complex growing environments (e.g., topography and landscape) make the area detection for late rice more challenging (Dong and Xiao, 2016; Liu et al., 2020). The following study could consider utilizing geostationary satellite observations to increase the temporal frequency of remote sensing data during the transplantation period of late rice in China (Shen et al., 2023). Subsequently, we will try to automate the generation of ChinaRiceCalendar based on the 'rgee' package (Aybar et al., 2023) and update the database once a year.'

5. The conclusion section must be improved. It currently looks like the repeat of abstract. Avoid repetition of information already presented in the abstract. Instead, focus on summarizing the findings and emphasizing the scientific contributions your work offers.

We modified the Conclusion section as follow (Lines 377-394):

'In the study, we improved the procedure of growing season division in the PhenoRice algorithm, and detected rice areas and rice phenology in early, middle, and late seasons across China from 2003 to 2020. Then, we established a multi-season rice calendar dataset named ChinaRiceCalendar. Firstly, the detected rice areas in ChinaRiceCalendar show a good agreement with statistical sown areas of rice in various growing seasons. The $R^2$ between the detected and statistical areas of ER, MR-SLR, and DLR at the province level is 0.92, 0.83, and 0.85, respectively. Secondly, the key phenological dates in ChinaRiceCalendar have high consistency with the field observations from 338 Agricultural Meteorological Stations in China. The RMSE between data from ChinaRiceCalendar and AMSs for rice transplanting, flowering, and maturity dates in China is 8.34, 7.84, and 7.77 days, respectively. Thirdly, ChinaRiceCalendar shows higher accuracy in the detected rice area or key phenological dates by growing season than RiceAtlas, ChinaCropPhen1km, and RICA in China. According to the calendar data from 2003 to 2020, the transplanting dates for early, middle, and late rice shifted by +5.4, +2.6, and -5.7 DOY/decade, respectively; the flowering date for early, middle, and late rice shifted by +5.5, -2.8, and -2.7 DOY/decade, respectively; the maturity date for early, middle, and late rice shifted by +3.2, -3.6, and -5.1 DOY/decade, respectively. Overall, ChinaRiceCalendar provides more reliable data to investigate and optimize the spatio-temporal structure of rice cultivation in China under climate and land-use change.'

6. The paper lacks a data sustainability plan, which is crucial for readers intending to reuse the data. The data only available through 2003-2020. What about the following years? Are there plans to continue the project in subsequent years? It is unfortunate that many projects and data were discontinued after the paper was published. I am looking forward a data management plan given the manuscript is submitting to a scientific data journal. For instance, if the project is to continue, what is the operational plan? If not, how can users reproduce the data independently?

Thank you for your questions, we added the following sentence into the manuscript (Lines 369-371):

'Subsequently, we will try to automate the generation of ChinaRiceCalendar based on the 'rgee' package (Aybar et al., 2023) and update the database once a year.'

**Reviewer #3:**

Major comments:

1. To align with the journal's standards, it is encouraged that the authors could enhance the clarity and coherence of the writing. For example, lines 88-91, the sentence, "However, we observed that the differences in rice types due to ignoring the actual cropping dates in previous studies made the current rice crop calendar focus on the key phenological dates in mono seasons (Luo et al., 2020; Qiu et al., 2017) and did not explore the spatial and temporal distribution for different rice varieties in China," seems to lack a clear subject in its second segment. This could potentially confuse readers and reduce the quality of paper.

2. The introduction section requires further refinement for clarity. The connection between the actual cropping dates and the challenges they present in understanding the spatial and temporal trends is not clearly elucidated. It is encouraged that the authors could provide a more detailed explanation or rephrase the statement to clarify why exactly these actual cropping dates introduce complications in analyzing the different rice varieties' trends.

3. Following the previous question, I'm curious about the rationale behind the categorization of "early," "middle," and "late" rice. What is the significance of segmenting the rice calendar into these three distinct stages? How does your methodology specifically aid in distinguishing between early, middle, and late rice? While there are mentions related to this, they are somewhat scattered and not easily discernible. There's a need for more comprehensive information on this topic. It would be beneficial for readers if these points were more cohesively presented.

Thank you for the comments. We have invited an expert to thoroughly polish the English language of the manuscript.

The three comments above indicate that we failed to convey the shortcomings of the existing rice calendar datasets and the significance of our study in the Introduction section. Therefore, we rewrote the shortcomings of previous rice calendar studies and clarified the significance of accurately dividing rice seasons in the Introduction section as follows (Lines 58-85):

'The PhenoRice algorithm excels at extracting rice phenology in multiple cropping systems and has been widely used in East Asia, South Asia, Southeast Asia, and Europe (Busetto et al., 2019; Liu et al., 2020; Mishra et al., 2021). However, the performance of the PhenoRice algorithm depends on the division of rice seasons, which requires expert knowledge about rice-based cropping systems in different regions (Mishra et al., 2021).'

'In China, there are at least three rice-growing seasons (early, middle, and late seasons) in diverse rice-based cropping systems (e.g., single-rice, double-rice, rice-wheat, rice-rapeseed, and rice-vegetable systems) (Frolking et al., 2002; Qiu et al., 2003; Cao et al., 2021; He et al., 2021). Generally, early-, middle-, and late-season rice are transplanted at Day Of Year (DOY) 30-130, DOY 110–180, and DOY 150-230, respectively. The growth periods after transplanting for early, middle, and late rice are 70-100 days, 100-130 days, and more than 130 days, respectively. Although field observations are important data sources for studying rice calendars in different growing seasons, they are usually limited by spatial and temporal discontinuities (Zhao et al.,

2016; Wang et al., 2017). Therefore, previous studies have typically utilized satellite remote sensing products to establish rice calendar datasets at the regional scale (Shihua et al., 2014; Liu et al., 2019; Bai and Xiao, 2020; Luo et al., 2020; Mishra et al., 2021). Nevertheless, these calendar datasets based on satellite remote sensing do not rationally classify rice growing seasons across China. For example, the dataset ChinaCropPhen1km only distinguishes between early and late rice in double-rice systems (Luo et al., 2020); the assumptions of the dataset RICA about rice flowering dates in different seasons do not correspond to the realities in China (Mishra et al., 2021); Shen et al. (2023) produced high-resolution distribution maps of single-season rice but did not explore multiple rice cropping systems. Early-, middle- and late-season rice in China are not only planted at different times, but also have distinguishing varietal characteristics, such as different temperature and photoperiod sensitivities (Zong et al., 2021). Thus, a crop calendar that accurately classifies rice seasons will provide reliable data for agricultural models to calibrate crop parameters at the variety level. Moreover, effective identification of different rice seasons will help analyze the response and adaptation of rice phenology to climate change.'

4. The manuscript proposed an "enhanced PhenoRice algorithm", yet a distinct clarification of the modifications compared to the original PhenoRice is missing. From the current description, it appears that the primary enhancement of the refined version seems to just add the prediction of maturity. It's essential for the authors to provide more detailed info about the modifications, explaining the rationale behind each and how they contribute to the study's objectives. This will offer readers a clearer understanding of the research's unique contributions and its potential implications.

5. The section titled "Extraction of rice growth information" (Line 194) lacks clarity and readability. As it currently stands, readers may need to refer to the original PhenoRice paper to understand the content fully. This is not ideal for an independent paper. Additionally, throughout the methodology section, various terms and methods, such as "wWHIT," are introduced without adequate context or explanations. To make this paper more reader-friendly and easy to understand, it's imperative for the authors to elaborate on these terminologies and provide a more thorough introduction to the methodologies employed. Specifically, highlighting the differences and innovations compared to previous works would greatly enhance the paper's clarity and value.

Thank you for the two comments above. We rewrote the section titled 'Extraction of rice growth information' and added more details of the improved PhenoRice algorithm in the Methods section (Lines 151-205):

**2.3.2   Estimation of rice area and cropping calendar**

[revised manuscript text omitted]

6. To more compellingly establish the merits or improvements of your methodology, it is encouraged that the authors could directly compare the accuracy of your product against ground truth data, along with similar accuracy evaluations from existing or competing products. Instead of solely contrasting the final datasets or only showing the alignment of your product with ground truth data, incorporating a multi-faceted comparison using ground truth as a consistent benchmark would provide readers with a good understanding of your research's relative strengths and contributions.

Thank you for your advice. In Section 3.2, we compared the detected rice phenological dates in ChinaRiceCalendar with the field observations from 338 Agricultural Meteorological Stations (AMSs) in China (Lines 241-259) .

Also, we compared the rice phenological dates obtained from the RiceAtlas (Laborte et al., 2017), ChinaCropPhen1km (Luo et al., 2020), and RICA (Mishra et al., 2021) datasets with the spatially corresponding AMS data. The comparison results are shown below (Lines 260-278):

**3.3 Comparison with other calendar datasets**

The rice phenological dates obtained from the RiceAtlas (Laborte et al., 2017), ChinaCropPhen1km (Luo et al., 2020), and RICA (Mishra et al., 2021) datasets were also validated against the spatially corresponding AMS data. In China, the rice phenological dates estimated in ChinaRiceCalendar show higher accuracy by growing season than those obtained from RiceAtlas and RICA. Furthermore, compared to the ChinaCropPhen1km dataset, the ChinaRiceCalendar dataset have similar accuracy in rice phenological dates but higher accuracy in rice areas in different growing seasons.

The RMSE between RiceAtlas' and AMSs' phenological dates for early, middle, and late rice in China is 18.27, 21.03, and 13.81 days, respectively. The $R^2$ between RiceAtlas' and AMSs' phenological dates for early, middle, and late rice in China is 0.65, 0.43, and 0.75, respectively.

The RMSE between ChinaCropPhen1km's and AMSs' phenological dates is 9.35 days for early and middle rice, and 7.24 days for late rice in China. The $R^2$ between ChinaCropPhen1km's and AMSs' phenological dates is 0.81 for early and middle rice, and 0.85 for late rice in China.

The RMSE between RICA's and AMSs' phenological dates for early, middle, and late rice in China is 22.80, 14.07, and 13.61 days, respectively. The $R^2$ between RICA's and AMSs' phenological dates for early, middle, and late rice in China is 0.47, 0.69, and 0.73, respectively.

(The RMSE between ChinaRiceCalendar's and AMSs' phenological dates for early, middle, and late rice is detailed in Section 3.2.)

7. Another study titled "RICA: A Rice Crop Calendar for Asia based on MODIS multi-year data" has also proposed a methodology rooted in the PhenoRice algorithm in 2021. Notably, their output map resolution matches yours at 250 meters. It's essential to discuss the differences and unique aspects of your method in comparison to theirs. Specifically, how does your study differentiate and enhance the existing body of work?

Firstly, we pointed out the problem of the RICA dataset in dividing rice growing seasons in China (Lines 77-78). Secondly, we improved the procedure of growing season division in the PhenoRice algorithm (Line 152-158). Thirdly, we compared the accuracy of ChinaRiceCalendar and RICA in rice phenological dates and found that the rice phenological dates estimated in ChinaRiceCalendar showed higher accuracy by growing season than those obtained from RICA (Lines 260-278).

Minor comment:

1. In Fig.3, there's a noticeable discrepancy in the results for the late cultivars when compared to other varieties. It is encouraged that the authors could clarify about this difference. Is it due to limitations in the methodology, inherent variability in the late cultivars, or other external factors? A clear explanation would assist readers in better understanding the data and its implications.

Thank you for your comments. Although $R^2$ between the detected and statistical areas of late rice is higher than 0.8 at the province level, there is still uncertainty in detecting late-rice area (Line 357-371):

'The uncertainty in crop area estimation is more significant for late rice than for early and middle rice, resulting in lower accuracy of the detected rice area in southern China (MLY, SC, SCS, YGP) than in northern China (NP, HP, LP). For example, there is an underestimation of the double-season late rice area in Hainan Province and an overestimation of the single-season late rice area in Hubei Province. Because the transplanting dates (DOY150-210) of late rice coincide with the rainy season in the main rice-producing areas, there is a higher risk of misidentifying agronomic flooding signals during transplanting for late rice than for early and middle rice. Furthermore, low data quality induced by cloud contamination during the transplanting period contributes to the difficulties in extracting the late rice area (Xiao et al., 2005; Clauss et al., 2016). Also, the diverse multi-cropping systems and the complex growing environments (e.g., topography and landscape) make the area detection for late rice more challenging (Dong and Xiao, 2016). The following study could consider utilizing geostationary satellite observations to increase the temporal frequency of remote sensing data during the transplantation period of late rice in China (Shen et al., 2023).'

2. In Figure 7, it is unclear what the content in the bottom right corner box represents. Clarifying this element could improve the reader's understanding of the data presented in the figure.

Thank you for your reminder. We modify the figure as following (Figure 6 in the new manuscript):

[Figure]

**Fig.6 Spatial distribution of rice areas in China during 2003-2020 (a: early rice, b: middle rice, c: late rice)**

3. For Figure 9, it would be beneficial to enhance the resolution for better clarity.

The Figure has been replaced with a clearer image (Figure 8 in the new manuscript).

[Figure]

**Fig.8 Rice phenological dates in main agricultural regions between 2003 and 2020 (a: Transplanting dates; b: Flowering dates; c: Maturity dates)**

4. On line 184, the term "weight" is mentioned but lacks a clear definition or context. This ties back to Major Comment 2, where various technical details are introduced throughout the paper without sufficient elaboration.

5. On line 188, the term "coarse fitting method" is mentioned, but there isn't a clear explanation provided for it.

Thank you for the two comments above. We rewrote the description of the fitting method (Lines 163-171):

The weighted Whittaker method in the phenofit R package was employed to smooth the MODIS-EVI time series (Kong et al., 2022). The Whittaker smoothing function can robustly capture seasonal signals with little noise interference, and it is widely used to identify crop phenology (Atzberger and Eilers, 2011; Bush et al., 2017). The curve fitting mainly relies on information from good-quality points, but also extracts the limited information available from the marginal- and bad-quality points. During the rough fitting to the EVI time series, we categorized the data quality of the observations according to their Quality Control (QC) information (SummaryQA of MOD13A1) and assigned weights of 1.0, 0.5, and 0.2 to the good-, marginal-, and bad-quality VI observations, respectively.

6. On line 220, there's an unnecessary space at the start of the paragraph that should be removed. Additionally, throughout the paper, there are instances where spaces seem to be missing. It's uncertain whether this issue arises from the original manuscript or due to PDF conversion. If the problem originates from the original version, it's recommended to address these formatting inconsistencies.

Thank you very much for the reminder. We have addressed the formatting problems.

7. On line 224, there appears to be a typo related to "R2." Please review the whole manuscript and correct all these typos to ensure accuracy and maintain the quality of the manuscript.

Thank you. We have double-checked the manuscript and corrected all typo issues.